# Bioactive Endodontic Hydrogels: From Parameters to Personalized Medicine

**DOI:** 10.3390/ijms241814056

**Published:** 2023-09-13

**Authors:** Marianne Leveque, Mourad Bekhouche, Jean-Christophe Farges, Audrey Aussel, Kadiatou Sy, Raphaël Richert, Maxime Ducret

**Affiliations:** 1Laboratoire de Biologie Tissulaire et Ingénierie Thérapeutique, UMR 5305 CNRS/UCBL, 69007 Lyon, France; marianne.leveque@ibcp.fr (M.L.); mourad.bekhouche@univ-lyon1.fr (M.B.); jean-christophe.farges@univ-lyon1.fr (J.-C.F.); 2Faculté d’Odontologie, Université Claude Bernard Lyon 1, Université de Lyon, 69008 Lyon, France; kadia2sy@gmail.com (K.S.); raphael.richert@insa-lyon.fr (R.R.); 3Service d’Odontologie, Hospices Civils de Lyon, 69007 Lyon, France; 4BIOTIS—Laboratory for the Bioengineering of Tissues (UMR Inserm 1026), University of Bordeaux, Inserm, 33076 Bordeaux, France; audreyaussel@hotmail.com; 5UFR d’Odontologie, Université de Bordeaux, 33600 Bordeaux, France; 6CHU de Bordeaux, Pôle de Médecine et Chirurgie Bucco-Dentaire, 33076 Bordeaux, France; 7Laboratoire des Multimatériaux et Interfaces, UMR CNRS 5615, Université Claude Bernard Lyon 1, 69622 Villeurbanne, France

**Keywords:** regenerative endodontic procedures, hydrogels, tissue engineering, apical release, preclinical model, review

## Abstract

Regenerative endodontic procedures (REPs) aim at recreating dental pulp tissue using biomaterials such as hydrogels. Their bioactivity is mostly related to the nature of biomolecules or chemical compounds that compose the endodontic hydrogel. However, many other parameters, such as hydrogel concentration, bioactive molecules solubility, and apex size, were reported to influence the reciprocal host–biomaterial relationship and hydrogel behavior. The lack of knowledge regarding these various parameters, which should be considered, leads to the inability to predict the clinical outcome and suggests that the biological activity of endodontic hydrogel is impossible to anticipate and could hinder the bench-to-bedside transition. We describe, in this review, that most of these parameters could be identified, described, and studied. A second part of the review lists some challenges and perspectives, including development of future mathematical models that are able to explain, and eventually predict, the bioactivity of endodontic hydrogel used in a clinical setting.

## 1. Introduction

Endodontics encompass a wide range of approaches which aim at diagnosing, preventing, and treating pulpal and periapical diseases and endodontic infections [1], including root canal treatment, one of the most performed dental procedures worldwide [2,3]. It involves filling the endodontic space with a sterile root filling material, gutta percha. However, since this filling material is bioinert, root canal treatments often result in the loss of the tooth’s immune defense, vascularization, and regenerative potential.

To address this limitation, researchers began exploring regenerative endodontic procedures (REPs) as early as in the 1960s to restore pulpal vitality. This led to the emergence of “revascularization” procedures in the 2000s, which are based on the concept of using patient-derived bioactive materials in the context of cell-homing within the cleaned endodontic space [4,5,6]. REPs are clinical strategies that aim at recreating dental pulp tissue using strategies such as tissue engineering. REPs are based on the replacement of the inflamed dental pulp by using a temporary scaffold based on hydrogels composed of biomacromolecules, potentially associated with cells, and called endodontic hydrogels in this paper. The context of the endodontic root is one of the highest challenges for REPs due to its size and anatomical complexity [7]. Numerous clinical and physicochemical parameters, such as apex size and endodontic shape, were reported to influence the outcome of REPs and results have varied greatly between patients [8,9,10]. Recent research in the field of dental materials and endodontics has increasingly focused its attention in developing innovative bioactive materials that can improve molecules’ release and promote tissue repair and regeneration [11,12,13]. Bioactivity, which can be defined as the biological activity of a device or drug, is mostly linked to the nature of the biomolecules and chemical compounds incorporated that can be released in the surrounding tissue [14]. Most of the endodontic hydrogels that were studied for REPs incorporated bioactive compounds such as antibiotics, peptides, or nanoparticles into their hydrogel, thus making them bioactive [13,15,16]. A great variety of bioactive hydrogels has been studied in the literature. An overview is proposed in Figure 1. Most are bioinspired, such as gelatin-based (GelMA), fibrin-based, or hyaluronic acid-based scaffolds [17,18,19,20,21,22,23,24,25,26,27,28,29]. Some studies used scaffolds that will react or dissolve at certain pH, or that are made from self-assembling peptides [19,25,30,31,32]. A few used microspheres or nanofibers to build a bioactive scaffold laden with cells or active molecules [31,33,34].

However, one of the major scientific and clinic locks for REP strategies is the lack of knowledge regarding the various parameters that could influence the release of bioactive compounds, especially because of the number of parameters that needs to be considered to predict and control such mechanisms [35,36]. This point is problematic because it will complicate the clinical validation and hinder the transition from bench to bedside. In this article, we reviewed most of the parameters that are or could be identified, individualized, and studied in the near future. The clear identification of these biophysical parameters will allow mathematical modeling to generate in silico tools to predicts bioactivity and its potential release.

This article proposes, first, to review and define what could be considered to be bioactive endodontic hydrogels, and, then, identify and discuss some parameters already investigated in the literature that could influence this bioactivity. Finally, the last section considers some challenges and perspectives regarding the future of this field of research.

## 2. Bioactive Endodontic Hydrogel

The bioactivity of a biomaterial refers to its ability to interact with biological systems such as living tissues. A variety of bioactive materials were already designed and proposed to improve interactions between cells and tissues, triggering processes such as cell adhesion, proliferation, differentiation, and the promotion of tissue regeneration [37,38]. Surface properties and chemical composition of these materials play a crucial role in facilitating biocellular interactions, allowing the materials to merge with the surrounding biological environment. Bioactive endodontic hydrogels could be considered to be advanced and innovative materials for REPs that were specifically designed to promote and enhance dental pulp regeneration.

Many endodontic hydrogels have already been suggested and reported in the literature for REPs, mostly in injectable forms [39,40,41]. The main components of bioactive endodontic hydrogels include a matrix, one or more bioactive agents, and a cross-linking mechanism. A hydrogel matrix provides a three-dimensional scaffold that can mimic the natural extracellular matrix, offering support for cell adhesion and tissue ingrowth. Several polymers, such as chitosan, gelatin, or hyaluronic acid, were commonly reported as promising materials for REP, due to their biological properties [18,19,42]. The cross-linking mechanism is also a key parameter to consider and can be achieved through physical or chemical methods [43,44].

The use of bioactive endodontic hydrogels in root canal therapy offers several advantages over conventional inert materials. Firstly, an endodontic hydrogel, which is a bio-inspired hydrogel such as a fibrin-based hydrogel, will be able to a promote regenerative process, characterized by the formation of a dentin-pulp-like complex [45]. Secondly, the bioactive agents incorporated in the hydrogel are easily released to create a favorable microenvironment for tissue regeneration. Bioactive agents could be biomolecules such as growth factors that stimulate tissue repair and regeneration. Growth factors, such as transforming growth factor-beta (TGF-β), bone morphogenetic proteins (BMPs), and vascular endothelial growth factor (VEGF), have already been reported to stimulate cell proliferation and differentiation, angiogenesis, and tissue regeneration of the dental pulp when incorporated into hydrogels [29,46,47,48]. Additionally, bioactive endodontic hydrogels may include antimicrobial agents, such as chlorhexidine or silver nanoparticles, to avoid bacterial infection and reduce the risk of treatment failure [39,49,50,51,52,53,54]. Furthermore, bioactive endodontic hydrogels can be clinically administered using minimally invasive techniques, such as syringe injection, allowing for precise placement within the endodontic space. This minimizes the risk of per operative complications and promotes the preservation of the remaining tooth structure.

In conclusion, endodontic hydrogels, notably those based on bio-inspired molecules such as fibrin or chitosan, are promising candidates that could improve the success of REPs. The most challenging obstacle with these devices is how to control the spatio-temporal bioactivity to create a specific microenvironment that ensures prolonged effects to enhance REP’s chances of success. However, this controlled bioactivity is nowadays overlooked or simplified in the context of endodontic hydrogel, suggesting the need to deeply investigate this multifactorial problem.

## 3. Bioactive Endodontic Hydrogel: A Multifactorial Problem (Figure 2)

### 3.1. Properties of Hydrogels

The physical and chemical properties of any endodontic hydrogel used for REPs should have controlled biochemical properties (release, swelling…) to achieve proper dental pulp regeneration. Several parameters such as porosity, mechanical strength, or biocompatibility of the hydrogel could strongly promote or reduce it [55,56]. Moreover, cell adhesion and proliferation were also reported to influence the release kinetics of various molecules by contracting the hydrogel and synthesizing a new extracellular matrix that also affects the cell–biomaterial relationship [55,57,58,59]. This is illustrated by the work of Jeon et al. (2006), which reports that the lower the fibrin concentration, the greater the release of FGF2, which is a growth factor correlated with cell adhesion [60]. Changes of the endodontic environment can be explained by numerous characteristics, such as the ability of the hydrogel to contract and modulate its shape, its degradation over the time, the efficiency of molecules entrapping and releasing, or swelling and volume variations when in contact with fluids [22,61,62,63,64]. In summary, many biological, physical, and chemical properties can influence the hydrogel’s behavior and the bioavailability of supplemented bioactive compounds.

### 3.2. Properties of the Biomolecular Building Block

Hydrogels are formed by the 3D organization of biomolecular building blocks, which will organize into fibers in combination with biomolecules entrapped between the fibers.

The added biomolecules play a major role during the release from the endodontic space to the periapical environment [65,66,67]. Parameters of the released biomolecules, such as size, shape, charge, and solubility in hydrophilic or hydrophobic media, play a critical role in their bioactivity and bioavailability, as they modify its ability to diffuse through and interact with the hydrogel and the surrounding host tissue [18,68]. Current hydrogels developed for endodontic treatment include, most of the time, molecules such as growth factors, antibiotics, and anti-inflammatory agents [29,69,70,71]. Several studies point out the importance of physicochemical studies focusing on detailing diffusion kinetics inside hydrogels and gels from a mathematic point of view [55,63,72]. In summary, the physicochemical properties of the molecule in combination with a specific hydrogel will strongly affect reciprocal interactions and greatly influence the release kinetics and therefore the bioactivity of endodontic hydrogels.

### 3.3. Periapical Composition (Cells/Medium/pH, Inflamation)

The composition of the periapical host tissues can also influence the release of molecules from an endodontic material (hydrogel) [65,67]. Each molecule, according to its own physicochemical properties, will have varying affinities depending on the tissue, hydrogel, pH, etc. [63,68]. Labille et al. (2007) explored the way Brownian movement (the random movement of molecules in a solution) is modified in a gel/solution interface, and how the interface itself influences the gel’s behavior and porosity. Leddy et al. (2004) described how hydrogels surface in contact with organic structures or liquids would swell and degrade at various rates according to the properties of both the scaffold and the environment [35,61,62,73]. The inflammatory response, in the periapical areas, was also reported to potentially influence the local pH and therefore the ionization state of reactive groups, thus modifying the solubility and, in turn, the release kinetics of molecules from the endodontic material [74]. In summary, endodontic hydrogels are not isolated, and their interactions with surrounding host tissues and liquids will strongly influence their rheological, chemical, and bioactive properties.

### 3.4. Hydrogel Degradation and Remodeling

The degradation of the hydrogel also needs to be considered in the release kinetics of the bioactive molecules over time [17,75]. For example, some hydrogels such as fibrin, GelMa, or agarose have been used as an endodontic material for their ability to be degraded, inducing the gradual release of bioactive molecules over time [21,28,62]. The degradation rate of the material can be adjusted, especially by modulating the hydrogel concentration and cross-linking, or by adding specific enzymes, to achieve the desired release profile. Other studies, on different matrixes, also used the fine-tuning of gelatin hydrogels to control the degradation rate and, thus, the release [29,56,73]. However, in the case of fibrin hydrogels, cell colonization will lead to matrix contraction, and although the fibrin matrix might be degraded, the production of a new matrix by cells leads, according to Leddy et al. and Lepsky et al., to a decrease in molecule release [57,61,76].

### 3.5. Root Anatomy and Composition

The periapical diameter of the tooth is one of the main parameters to be considered as it could vary greatly in size, shape, and direction according to the patient and also influence the release properties accordingly [77,78]. Fick’s law of diffusion supposes a linear impact of the contact surface and concentration gradient on diffusion rates and amounts. The apical diameter determines the contact surface between endodontic hydrogel and host periapical tissues. As a result, the increase in the apical diameter should lead to an increase in the diffusion of the hydrogel components in the host periapical tissues. Fick’s law was validated in previous studies of Abbott et al. and Robert et al. [70,79], but subtle variations in the shape and the direction of the endodontic space still need to be taken into consideration. Moreover, it is well known that the bioactivity of endodontic hydrogel will not only rely on the periapical region but also on the diffusion of molecules across the porous surface of dentine around the root [80]. In the other way around, the release of molecules from dentine through the hydrogel is also a key point to considered. Indeed, dentin-derived growth factors were reported to be sufficient to promote the recruitment and differentiation of stem/stromal cells [78,81,82]. As a result, the overall diffusion through the hydrogel is a sum of bi-directional mechanisms of the release of hydrogel-derived molecules from the hydrogel to the dentin and of the release of dentin-derived biomolecules from the dentin to the hydrogel. The patient-specific shape of the endodontic space plays a crucial role in these mechanisms of release by varying the dentin–hydrogel contact surface. These points strongly complexify the modeling of the release of hydrogels and they suggest the need to integrate these anatomic parameters in order to personalize REP treatment.

### 3.6. Use of Carriers

Carriers or vectors can be used to deliver the therapeutic molecules to the site of interest in a controlled manner [50,53]. These carriers can be incorporated into the material structure or added separately as a coating or filler. Jeon et al. (2006) encapsulated bFGF on polymeric nanoparticles to achieve a controlled and sustained release over weeks and to limit the burst effect that was frequently observed in fibrin hydrogels [60]. Bekhouche et al. (2020) loaded clindamycin on PLGA nanoparticles to achieve a sustained release and maintain the antibacterial action within a fibrin hydrogel [27]. By modifying the vector’s properties such as ionic charge, hydrophilicity, etc., it is also possible to deeply modify the release kinetics from a material and to achieve finely-tuned bioactivity. Vectors could be cell- or space-specific using surface targeting or they could achieve an on-demand release in “smart” devices [55,83,84]. Figure 2 is an overview of all the parameters cited herein.

## 4. Bioactive Endodontic Hydrogels: Challenges and Perspectives

### 4.1. Deep Characterization of Biochemical Characteristics of the Endodontic Players

Bioactivity can be defined as the biological activity of a device or drug. Fibrin hydrogels are an example of potential endodontic hydrogels that harbor several bioactivities [20,21,23,24,27]. Hydrogel macromolecular properties such as stiffness, fiber and pore sizes, and reactive groups were reported to impact the bioactivity [25,71]. One can easily imagine the immense amount of data and knowledge necessary to achieve a deep understanding of the bioactive properties of a hydrogel, especially in combination with active molecules such as growth factors or antibiotics. This use of antibiotic components remains, however, controversial due to potential bacterial resistances and the lack of knowledge surrounding exact root canal delivery and apical diffusion to the periodontal host tissues leads to such components being avoided in the most recent guidelines on regenerative procedures [85]. Controlled delivery with a vector, such as a nanoparticle for example, adds a great number of parameters, such as size, solubility, or shape, that complexify the understanding of the release even further and, therefore, the bioactivity of the loaded bioactive molecule. Although the physicochemical characteristics of biomolecules are often well documented in the literature, they need to be taken into consideration in the specific context of encapsulation in a hydrogel, as well as for the patient-specific morphology of the endodontic space to control the in situ and released bioactivities [57,65,66,67,86]. Encapsulating an active molecule in a hydrogel makes the hydrogel a de facto vector, and the release of an active molecule with a specific release profile causes them to become not only scaffolds but also galenic objects.

Galenic could be defined as the science of a medication’s shape and vector, which could be optimized according to the molecule vectorized, as well as the patient’s context and needs. In order to safely go from bench to bedside, hydrogels in endodontics must be explored as galenic objects, as well as scaffolds, and it would be interesting to conduct more galenic research and scale-up investigations to explore the bioactivity of these devices.

### 4.2. Create and Validate Standardizable Models

The study of endodontic hydrogel bioactivity will require, in the future, simple, cheap, and reproducible models to study all the previously listed parameters and therefore improve our knowledge. From a methodological point of view, the release of molecules from the apex was studied for decades with “ex vivo” models, in vitro models such as pipettes hung in syringes, and in vivo models such as roots implanted in mice backs [36,79,80]. However, the “ex vivo” model using human teeth is challenging due to the difficulty, or even impossibility, of obtaining human tissues. Moreover, it involves labor-intensive and time-consuming manual preparation of each endodontic space, and the interindividual variability in root and canal anatomy can lead to heterogeneous results and masks the effect of each parameter. To overcome these limitations, an alternative model was proposed using a pipette system hung in a syringe filled with agarose to simulate the apical environment and study the release of Ca(OH)_2_. Although this new model eliminated the need for human tissues, it still required significant preparation time for each individual pipette [79]. Finally, a root model implanted in a mouse was used to explore the release of radio-traced molecules and gain a deeper understanding of the systemic consequences of endodontic materials releasing molecules in the whole organism. This model is interesting but also presented several limitations, as they are painful experiments for the animals and also comes with a high financial toll. This explains why most of the release kinetics of endodontic hydrogels are investigated with partial models, such as covering a hydrogel disk of PBS in a well or a beaker. This approach provides a relatively large surface area, sometimes exceeding 1 cm^2^, which is unrealistic for assessing apical release and can lead to an overestimation of any potential clinical effects. This observation is supported by study results showing a maximum of a 35% release after 24 h from fibrin in an endodontic model, while Lepsky et al. (2021) reported up to a 90% release at the same time point, with a partial model. This dramatically overestimates the potential bioactivity of any endodontic hydrogel [28,32,34,87]. To address this issue and achieve a more standardized approach for investigating apical release, a simple and efficient model can be suggested.

### 4.3. Emulate Mathematical Model for Understanding and Personalize Endodontic Hydrogels (Figure 3)

Descriptive mathematical models, also called fitting models, such as the Peppas–Korsmeyer or the first-order kinetic model, have already been used to investigate the release from an endodontic biomaterial [28]. Dubey et al. used a model to fit and describe the release of antibiotics from a GelMa endodontic hydrogel and concluded the gel’s release kinetic as being a mix of Fickian diffusion and swelling-triggered diffusion in PBS [28]. Peppas, as well as other curve fitting models are powerful tools for describing the diffusion from a hydrogel and they contribute greatly to a better understanding of the behavior of bioactive materials [28,88,89].

In the past decade, there has been a remarkable surge in the generation of experimental and clinical data in biology and dental medicine. However, due to the wide diversity of models used, it has become challenging to compare and extrapolate findings. The future advancements in artificial intelligence and 3D imaging of endodontic spaces hold the potential to address this issue by designing virtual models based on scientific databases, such as those produced by omics for high throughput screening and the prediction of pathological biomarkers [90,91]. Similarly, such approaches could be efficiently used to anticipate the bioactivity of these hydrogels based on galenic properties of endodontic hydrogels [88,89]. In the future, this would lead to an even more personalized medicine, which is tailored to each patient’s needs and biological parameters, such as age, inflammation, bacterial load and strains, root anatomy, or medical history [92]. Figure 3 proposes a synthesis of this perspective.

## 5. Conclusions

Anticipating the bioactivity of a hydrogel in and around the periapical area is complex, mostly due to the numerous parameters that could influence the release of molecules. Understanding these properties can help guide the development of endodontic hydrogels that can effectively release therapeutic molecules and drive a more personalized bioactivity.

To achieve this goal, numerous scientific and technical obstacles still exist. Further studies are required to: (1) improve the knowledge of the release of molecules and endodontic hydrogels, to ensure knowledge and safety regarding all products brought to endodontic practice, (2) create and validate standardizable models to enable investigation of a greater number of parameters influencing the release in a more precise and efficient manner, (3) emulate mathematical models that could predict the release from any endodontic hydrogel and as a result anticipate the behavior of hydrogels in a clinical setting. This point would allow us to go directly from in silico modeling to clinical use in a safe and controlled manner.

## Figures and Tables

**Figure 1 ijms-24-14056-f001:**
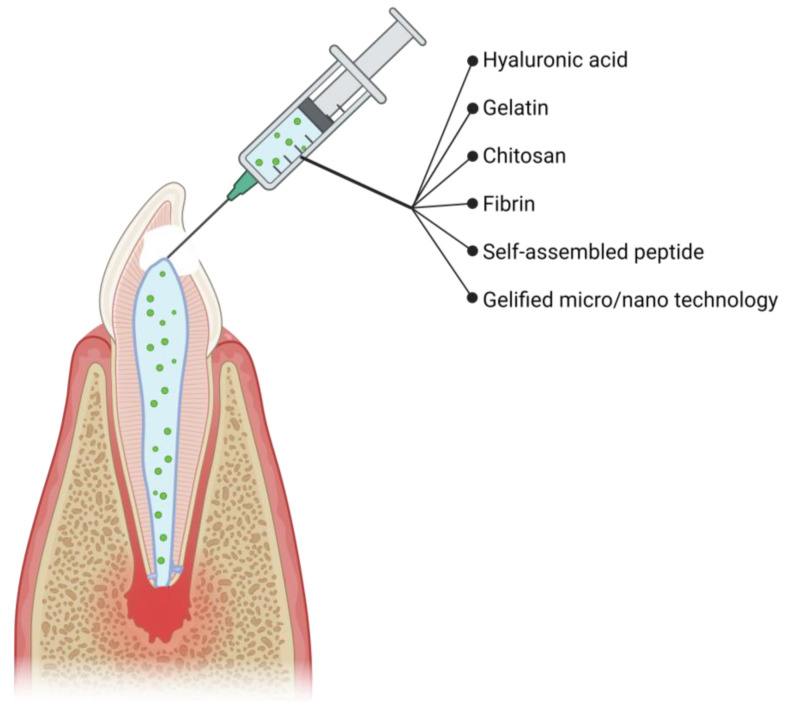
Overview of most bioactive endodontic hydrogels that have been published [17,18,19,20,21,22,23,24,25,26,27,28,29,30,31,32,33].

**Figure 2 ijms-24-14056-f002:**
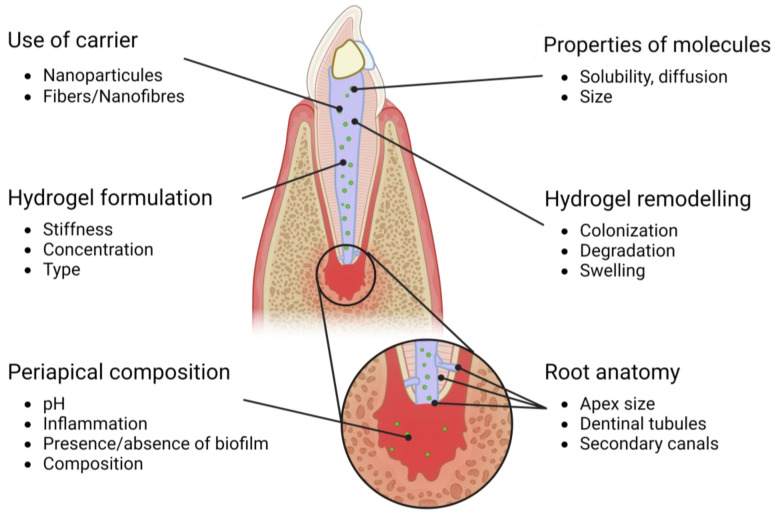
Overview of parameters that could influence the bioactivity of endodontic hydrogels.

**Figure 3 ijms-24-14056-f003:**
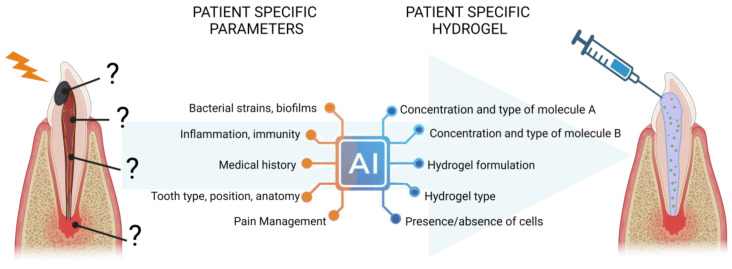
Perspectives of personalized endodontic hydrogel using AI models to design the most adapted device according to patient-specific parameters.

## Data Availability

Data sharing not applicable—no new data generated.

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
