# Peer review of "Bioactive Endodontic Hydrogels: From Parameters to Personalized Medicine"

_ijms, 2023, doi:10.3390/ijms241814056_

Round 1

Reviewer 1 Report

This paper is a very good review of the past problems and future possibilities of Tissue Enginnering with Hydrogel for Root Canal treatment.   Two thing I would like to request is

1)  to provide a comprehensive table of the materials and outcomes of the experiments on the Hydrogel papers that have been published so far.

2) Figs of the actual material and how the Hydrogel is placed in the Root canal would also be very helpful to the reader, so I would recommend adding Figs.

Author Response

  • To provide a comprehensive table of the materials and outcomes of the experiments on the Hydrogel papers that have been published so far.

RESPONSE:

We thank the reviewer for this comment. Introduction has been modified to include all the endodontic hydrogels identified in the literature that investigated the release of a molecule.

We considered the question of a table, but very few studies explore release from an endodontic hydrogel, and even less explore apical release in itself. There is also a great variety in the protocols and results, which led us to believe a table would be more confusing than enlightening for the reader. This heterogeneity is the reason why we chose a narrative review instead of a scoping or systematic review.

Figure 1 was added to the Introduction section, listing materials found in the literature so as to provide more comprehensive information. However, if the Reviewer or the Editor strongly believes that a table would improve the quality of this manuscript, we can generate a synthetic or comprehensive table.

  • Figs of the actual material and how the Hydrogel is placed in the Root canal would also be very helpful to the reader, so I would recommend adding Figs.

RESPONSE:

We thank the reviewer for this comment. We emulated the following additional figure to inform the reader of endodontic hydrogels handling and simultaneously provide an overview of the different hydrogels available in the literature:

Reviewer 2 Report

Very good review manuscript with proper scientific structure. Clearly stated aims and concisely reviewed current literature. This manuscript will bring something new to the existing literature. Technical aspects of regenerative endodontic materials were discussed, and new potential areas of research were identified. These would facilitate future research and thus paper has a potential for citations. Personally, I really like concise and compact reviews such as this paper. Suitable for publication if authors correct following minor issues:

·      Lines 138, 151, 165, 222-235 – citations are missing. Please cite papers that support statements included in those paragraphs. 

·      Ln 205- 206 – Words in brackets are not proper form of scientific writing. Please correct.

·      Conclusion section is scare. It should be expanded. Authors should briefly explain why these points (ln 294-297) were selected. Readers should be convinced of importance of identified research areas. 

Author Response

  • Lines 138, 151, 165, 222-235 – citations are missing. Please cite papers that support statements included in those paragraphs.

RESPONSE:

We thank the reviewer for this comment. The following citations were added to support our claims

Line 138 : Iohara et al, 2023, Zhang et al, 2021, Liu et al, 2016

Line 151 : Iohara et al, 2023, Lui et al, 2016, Stylianopoulos et al, 2010, Sultanova et al, 2016

Line 165: Trusek et al, 2021, Silva et al, 2018

Lines 222 – 235: Liu et al, 2016, Zhang et al, 2021, Iohara et al, 2023, Lepsky et al, 2021, Hasan et al, 2007

  • Ln 205- 206 – Words in brackets are not proper form of scientific writing. Please correct.

RESPONSE:

We thank the reviewer and modified the text accordingly:

By modifying the vector’s properties such as ionic charge, hydrophilicity… It is also possible to deeply”

  • Conclusion section is scare. It should be expanded. Authors should briefly explain why these points (ln 294-297) were selected. Readers should be convinced of importance of identified research areas.

RESPONSE:

We thank the reviewer for this comment and agree. Conclusion was modified in accordance:

Anticipating the bioactivity of a hydrogel in and around the periapical area is complex, mostly due to the numerous parameters that could influence the release of molecules. Understanding these properties can help guide the development of endodontic hydrogels that can effectively release therapeutic molecules and drive a more personalized bioactivity.

To achieve this goal, numerous scientific and technical locks still exist. Further studies appear required to: 1) improve knowledge of the release of molecules and endodontic hydrogels, to ensure knowledge and safety regarding all products brought to endodontic practice, 2) create and validate standardizable models to enable investigation of a greater number of  parameters influencing the release, in a more precise and efficient manner, 3) emulate mathematical model that could predict release from any endodontic hydrogel and as a result anticipate hydrogels behavior in clinic. This point would allow to go directly from in silico modeling to clinical use in a safe and controlled manner.”
